# Hemostatic Efficacy of Absorbable Gelatin Sponges for Surgical Nail Matrixectomy after Phenolization—A Blinded Randomized Controlled Trial

**DOI:** 10.3390/jcm11092420

**Published:** 2022-04-26

**Authors:** Antonio Córdoba-Fernández, Adrián Lobo-Martín

**Affiliations:** Departamento de Podología, Universidad de Sevilla, C/Avicena s/n, 41009 Sevilla, Spain; adrlobmar@gmail.com

**Keywords:** ingrown toenail surgery, hemostatic gelatin sponges, bleeding, phenolization

## Abstract

Background: Some studies have recommended combining germinal matrix excision with phenol ablation in the treatment of onychocryptosis. Matrixectomy after phenolization has been shown to be an effective modification to reduce the drawbacks associated with phenolization alone, although it increases the risk of minor postoperative bleeding. The present study aims to assess the effectiveness and safety of gelatin sponges as hemostatic agents in partial matrixectomy after phenolization. Methods: A comparative clinical trial in parallel groups was designed in 74 halluces (44 patients) with stage I, II, and III onychocryptosis. All participants were randomly assigned to 3 groups: Group A (control group), Group B (conventional gelatin sponge), and Group C (high porosity gelatin sponge). Results: The quantified mean blood loss in the first 48 h after surgery in patients in both experimental groups was significantly lower compared to the control group. The lowest mean blood loss was recorded in Group C (*p* < 0.001) and followed by Group B (*p* = 0.005). No adverse effects were recorded in any of the patients included in the experimental groups. Conclusions: Hemostatic gelatin sponges were demonstrated to be effective and safe devices for the control of minor postoperative bleeding associated with matrixectomy after segmental phenolization.

## 1. Introduction

Currently, segmental phenolization (SP) is the gold standard in the surgical treatment of onychocryptosis due to its low recurrence rate with a favorable adverse effect profile [1]. Studies that have evaluated Wedge resection (WR) matrixectomy combined with posterior SP have been demonstrated to be effective treatments with significantly fewer recurrences than the procedures used SP or WR alone [2,3,4]. However, SP or the combined procedure WR/SP appears to increase the risk of postoperative swelling and discharge with the consequent increased prolonged healing time and risk of infection related to the destruction of cauterized tissue [1,5,6]. The curettage or WR of cauterized tissue after SP has been demonstrated to be an effective modification of combined techniques to reduce the drawbacks associated with SP alone, but with an increased risk of postoperative bleeding and/or pain [7].

Minor postoperative bleeding after partial nail matrixectomy has been an item poorly analyzed as the primary outcome in reported clinical trials. Experimental studies that have compared SP or WR alone versus WR combined with posterior SP have demonstrated that postoperative bleeding was less after SP alone or in combination with WR [7,8,9]. Postoperative toenail wounds involve hemodynamic conditioning originating from the peripheral effect of the heart in a particularly vascularized zone such as the matrix and nail bed. This condition is exacerbated after surgery by reactive hyperemia produced on the toe after removal of the tourniquet, which increases in the standing position. Thus, achieving postoperative hemostasis after nail surgery is crucial due to the rich vascular supply of the nail bed and matrix. Platelet-rich fibrin has been shown to be effective in controlling postoperative bleeding and recovery time after SP or partial WR [10,11]. However, this turns out to be less cost-effective than other approved hemostatic agents, as the preparation of autologous blood products is complex and time-consuming with an increase in operative durations.

Most mechanical hemostats are based on animal-derived products, such as collagen and gelatin, especially purified gelatin derived from porcine skin. The advantage of these hemostatic devices is that they act not only by mechanical pressure but also by their physiological action mechanism, i.e., absorb blood cells and activate platelet aggregation, release coagulation factors, and activate endogenous hemostasis [12]. The most commonly used absorbable gelatin hemostatic agents are presented in sponge forms and have been demonstrated to be useful in different surgical specialties. In dermatologic surgery, gelatin sponges have been used to control postoperative hemostasis from capillary, venous, and low-pressure arteriolar bleeding [13]. 

Most collagen-derived products, such as gelatin sponges (GS), possess remarkable coagulation functions due to their good porous structure and hygroscopic properties. These properties have attracted the attention of most biomedical researchers, and include excellent biocompatibility, good biodegradability, cell interactivity, non-immunogenicity, and excellent processability, availability, and cost-effectiveness. Their high swelling capacity and rapid hemostatic ability make them suitable for preventing exudate accumulation and protecting the wound bed from bacterial invasion [14]. Due to its pH neutrality, GS work as an ideal drug carrier and can be very useful in combination with antifibronolitics or other hemostatic agents [15]. Absorbable GS soaked in tamponed aluminum chloride has been used successfully to achieve hemostasis after nail biopsy [16]. 

GS are flexible materials with well-interconnected micropore structures, with a pore size between 10 and 100 μm in diameter and interconnected channels. High-porosity gelatin sponges (HPGS) are characterized by high pore density, reduced linking, and high nanoscale with roughness of the lamella surfaces that have shown rapid hemostasis in vitro and in vivo models versus conventional gelatin sponges (CGS) [17]. Some animal model and clinical studies have demonstrated faster re-absorption of HPGS and a less inflammatory response than CGS, which produces less overall mass, resulting in a reduced risk of aberrant fibrosis, in addition, making it unnecessary to remove the dressingdue to its rapid biodegradation [18,19,20].

Currently, there are no studies that have analyzed the efficacy and safety of absorbable GS in nail surgery. We consider that SP and subsequent WR using GS can reduce these drawbacks, maintaining the advantage of combining procedures. Consequently, the objective of this experimental study is to evaluate the clinical efficacy and safety of two different types of GS after combining SP with the posterior WR matrixectomy. 

## 2. Materials and Methods

### 2.1. Study Design and Sample

A prospective single-center, parallel groups, randomized, double-blind study was designed. We recruited patients with ingrown toenails in hallux treated in the surgical section of the Podiatric Clinical Area of the University of Sevilla (Spain). The study was developed in accordance with the Consolidated Standards of Reporting Trials (CONSORT) guidelines. The inclusion criteria were hallux nail onychocryptosis stage I, II, or III according to the Kline classification [21] which indicated surgical treatment. Patients with erythema, drainage, and acute pain received conservative treatment prior to enrollment in the study. Patients with severe medical comorbidities (anemia, cardiovascular disease, uncontrolled diabetes, coagulation disorders, or patients with abnormal platelet count or taking antiplatelet aggregates or other oral anticoagulants) were excluded. The participants gave their written consent according to the Declaration of Helsinkiand the research protocol was approved by the Research Ethic Committee of the University hospitals Virgen Macarena and Virgen del Rocío (ID: 1206-N-15). The study was registered in the ClinicalTrials.gov PRS Registry (ID: NCT05140161).

All participants were randomly assigned to 3 groups using a simple equal-probability randomization scheme: Group A (control group), Group B (CGS group), and Group C (HPGS group). The final sample consisted of 44 participants with hallux-nail onychocryptosis (74 toes; 148 nail folds). The same researcher generated the random allocation sequence, enrolled participants, and assigned interventions. The investigator blinded each patient to the surgical procedure. The flow chart of the patients throughout the course of the study can be observed in Figure 1.

### 2.2. Surgical Procedure

For the three groups, the surgical procedure consisted of a partial nail avulsion associated with SP and posterior WR, as described by Winograd [22]. After a hallux nerve block with approximately 3–4 mL of 2% mepivacaine without vasoconstrictor, hallux blood was exsanguinated with a small Esmarch-type bandage and a soft rubber tourniquet was applied at the base of the toe. All procedures were performed on the medial and lateral nail folds of one or both halluces. A partial plate nail avulsion and posterior SP were performed using a 90% phenol solution that was applied for one minute with sterile gauze. After this, remnant phenol was dissolved in the area using 70% alcohol irrigation and then physiological saline solution. Subsequently, the cauterized tissue with a whitish appearance and granulation tissue, if present, was carefully removed using a scalpel and curette. The excised area included approximately 3–4 mm of the adjacent nail and nail bed down to the periosteum (Figure 2). The patients in each group were subjected to three different experimental conditions. The subjects of the control group (Group A) did not receive any treatment, while the patients in the two experimental groups received CGS (Octocolagen^®^, Clarben SA, Madrid, Spain) or HPGS (Gelita-Spon Standard^®^, Gelita Medical, Eberbach, Germany), respectively. Gelatin sponges were previously soaked in saline solution and prepared for their application by a different researcher from the one who performed the procedure. CGS was applied to the subjects of Group B and HPGS was applied to the subjects of Group C. Both types of sponges came in cubes of 10 × 10 × 10 mm size and were easily packed into the operated nail grooves covering the subeponychial space and the injured nail bed previously soaked with saline solution (Figure 3). All surgical wounds were covered with a non-adherent 6 × 4 cm sterile polypropylene dressing size (Apodrex^®^, Vectem SA, Barcelona, Spain) and five hydrophilic cotton gauzes (Tegosa SA, Toledo, Spain) were placed around the hallux and partially covered with a cohesive conforming bandage of size 400 × 4 cm (Peha-haft^®^, Hartmann, Heidenheim, Germany). The tourniquet was then removed and the bandage was finalized while the patient remained with the foot elevated (tremdelemburg position) for ten minutes before standing up.

### 2.3. Outcome Measurement

The patients were blinded for the treatment applied and the surgeon for the treatment applied to the patients in the experimental groups. The halluces of the control group received standard treatment with only non-adherent dressing, gauzes, and a compressive bandage while in the toes of the experimental groups in every operated groove a wet cube of the selected gelatin sponge of size 10 × 10 × 10 mm was also applied. Non-adherent dressings and five gauzes were previously weighed using a precision electronic balance (Nahitaserie 5152, d = 0.01 g), obtaining an exact of 4.29 g. For the analysis of postoperative inflammation, digital circumference (in cm) was measured using a flexible rule at the level of the proximal fold of the nail. The healing of the spontaneous wound closure was monitored by clinical evaluations and digital photographs. After 48 h, all participants returned for regulated and standardized dressing changes. Participants were reviewed 48 h after the surgical procedure patients and the elastic compressive bandage was removed. Non-adherent dressings and five gauzes were carefully removed together and subsequently weighed (Figure 4), and the digital circumference was again measured. In the control and experimental groups, the wounds were cleaned with a saline solution and 10% povidone iodine antiseptic solution was applied. In the experimental groups, when necessary, excess amorphous fibrin and gelatin sponge was partially removed with Adson forceps and the digital circumference (in cm) was measured again. The toe was covered with the same non-adhesive dressing, gauze, and elastic bandage. From the fifth day, patients were seen approximately every 48–72 h until the recovery period was complete. To limit subjectivity in the assessment of complete recovery time, clinical indicators of recovery time were considered when there was no drainage (no exudate evident), when the granulation tissue was covered by a scab (no evidence of granuloma or encapsulation), when there were no signs of erythematous tissue without evidence of infection, and the patient was able to use normal footwear and perform activities/work. All criteria had to be met before recovery time was reached. On day 5, when participants presented for redress, an experienced blinded clinician in nail wound care evaluated the wound. The recovery time was the interval between the application of the first dressing (at the time of surgery) and the clinical indicators were completely achieved. To measure postoperative pain, an analog visual scale for the self-evaluation of pain on a 10 cm scale (0 = absence of pain to 10 = unbearable pain) was used. Postoperative analgesia was administered with 500 mg of acetaminophen per os every 6 to 8 h (no more than 4 g/day) when pain measured with the chromatic scale was less than 5 and 1000 mg when it was more than 5. The same clinician performed all surgical procedures, and all parties involved in the postoperative period were blinded, including resident podiatrists, who collected the pain questionnaires. After a minimum follow-up of 8 months, an attempt was made to contact all patients for a telephone interview to assess satisfaction with the procedure. An independent and blinded evaluator conducted the telephone interviews. Recurrence was based on the presence of recurrent nail spicules or an ingrowing toenail with a minimum follow-up period of 8 months. Satisfaction with the procedure was analyzed on a 0–10-point scale.

### 2.4. Sample Size Calculation

The sample size calculation was performed with GPower 3.1.9 software (Universität Kiel, Kiel, Germany) based on a previous pilot study that investigated the postsurgical bleeding difference between partial matrixectomy after phenolization with or without the GS used in Group B. According to this pilot study, the percentage of abundant bleeding observed was 91.2% in the toes of the control group. Taking into account a clinically important reduction of 50% for this percentage, an error of 0.05 with a desired power of 80% (β = 20%) and a minimum sample size of 20 halluces per group, was considered. Taking into to account potential protocol violations, the research included additional toes in each of the experimental groups (5 in Group B and 9 in Group C). 

### 2.5. Statistical Analysis

Quantitative data were described as mean, standard deviation (SD), and 95% confidence interval (CI; lower and upper limits). For the analysis of the qualitative data, the Chi-square test was used to analyze the dependency relationship between the variables through cross tables. For the analysis between a categorical and a quantitative variable, normality tests were performed using the Shapiro–Wilk test to determine the most appropriate test based on the behavior of the data. To compare independent samples when the variables’ values met the normality criteria, the T test was used for two groups or ANOVA for three groups. When the variable to be analyzed did not meet the normality criteria, the Mann–Whitney U-test was used for two groups or the Kruskal–Wallis test for three groups. For related samples where the values of the variables were in accordance with normality, the *t*-test was used. To compare more than two related groups when the variable to be studied did not meet the normality criteria, Friedman’s two-dimensional analysis of variance by ranges was used.IBM SPSS Statistic software (v25, SPSS Inc., Chicago, IL, USA) was used for the data analysis and statistically significant differences were established at *p* < 0.05 with 95% CI.

## 3. Results

### 3.1. Descriptive Dates

A total of 52 patients (80 halluces) were included in the study conducted between March 2017 and November 2021. The final study sample consisted of 44 patients (74 halluces), of whom 19 were male and 25 were female. All participants were randomly assigned to three groups. The distribution of the three groups was homogeneous (*p* > 0.05), with respect to age variables (*p* = 0.94), sex (*p* = 0.60), nail morphology (*p* = 0.38), and stage of onychocryptosis (*p* = 0.35). The comparison between the laterality of the affected toes (right and left) used for the comparison between the groups demonstrated statistically significant differences (*p* < 0.05). The descriptive characteristics of the study sample are represented in Table 1.

### 3.2. Outcome Measurements

The outcome measures for each of the treatment groups are shown in Table 2. Regarding the main variable (blood loss), the Kruskal–Wallis test for independent samples demonstrated significant differences with respect to mean blood loss 48 h after surgery in patients from both experimental groups with respect to the control group. The most significant difference from the control group was recorded with respect to mean blood loss in Group C (*p* < 0.001) and followed by Group B (*p* < 0.005). No significant differences were found between the experimental groups (*p* > 0.999). The average difference in recovery time and the number of postoperative cures required between the control group and the experimental groups was not significant (Table 3 and Table 4). The evolution of pain measurements observed on the VAS scale during the first 3 days after surgery was similar in the three groups (Figure 4). The *t*-test for the related samples did not show significant differences in mean pain at 3 days postoperatively (Figure 5). The two-dimensional Friedman variance estimate by rank for related samples showed that the mean differences in pain at 3 days postoperatively did not differ significantly between groups. The *T*-test for related simples demonstrated that in the three groups there was a significant increase (*p* < 0.001) in their mean values between the mean digital circumference before and after surgery. The ANOVA test for independent samples did not show significant differences between the average digital circumference between the groups for recovery time and the number of postoperative cures required (Table 3 and Table 4). 

Out of 44 patients, 38 (86.36%) responded to the telephone interview. One hundred and thirty-four procedures (67 hallux) were analyzed, observing 10 recurrences, 4 asymptomatic (spicules), and 6 symptomatic, of which only three patients required a second operation. The satisfaction rate was 92.10%. The Mann–Whitney U-test for independent samples showed significant differences in the degree of satisfaction measured on a scale of 0–10 between the group of patients with or without recurrence (Table 5).

In any of the groups, complications were recorded with respect to the rate of postoperative infection, hypergranulation, encapsulation, or tissue reaction. 

## 4. Discussion

In the standard surgical approach for the management of ingrown toenails, matrix excision should be selective to minimize damage to the surrounding normal structures, but at the same time must be complete and reliable to prevent recurrences. Phenol is an effective protein denaturant that exhibits its cauterizing effect by producing a necrosis of coagulation in soft tissue with a higher incidence of postoperative discharge, hemorrhage, and risk of infection [1]. On the other hand, in chemical matrixectomy, regulation of the level of tissue destruction is uncontrolled and can result in bone injury [23]. The results of our study demonstrate that SP and subsequent WR using GS can reduce these drawbacks, maintaining the advantage of combining procedures.

We used the original WR described by Winograd in 1929, and the wounds were left open for secondary healing. In most studies that have combined both procedures, WR was performed prior to SP and the authors claim to use the technique described by Winograd, but the truth is that in most of them, the nail folds were constructed with the help of sutures [2,3,9]. In the original Winograd procedure, the author describes a small incision in the soft tissue of the nail fold and the eponychium in line with the toenail incision; the nail is cut, the ingrown portion is removed, and the matrix and nail bed are destroyed using a curette to prevent recurrence. The author noted that it was unnecessary to excise hypertrophic folds. The wound is left open for secondary healing and dressing changes are performed until the incision heals [22].

In our experience, the main associated drawback of WR alone or after SP is postoperative bleeding, which causes discomfort to the patient and sometimes requires more postoperative monitoring. Postoperative bleeding after nail surgery has been poorly studied in clinical trials, most often as a categorical variable [7,10,11]. Most nail surgery procedures, including SP, are usually performed with strict surgical ischemia using a tourniquet, as the accumulation of blood pooling in the nail bed is undesirable, makes the excision technically more difficult, and can dilute phenol below its optimal concentration. Although SP can reduce postoperative bleeding, when the tourniquet is removed, reactive hyperemia is produced in the hallux, which increases in the standing position with the consequent risk of postoperative bleeding and rapidly decreases the anesthetic effect, which can compromise patient welfare during the postoperative period.

The results of our study demonstrate that both CGS and HPGS are effective in reducing bleeding after combining SP/WR. Quantified mean blood loss in the first 48 h after surgery in the patients in the control group was significantly higher compared to both experimental groups. Four clinical trials have analyzed the combined efficacy of both techniques versus SP alone [2,3,7,9]. In two of them, greater bleeding was observed in the groups where both techniques were combined [7,9]. However, the results of these studies cannot be objectively compared with ours since bleeding was analyzed as a categorical variable and in the two of them in the combined procedure groups, nail folds were constructed with the help of sutures [3,9].

Some in vivo studies demonstrate that GS can act as a scaffold to support short-term cell survival and high-level growth factor production, exhibiting good clinical potential to improve wound healing [24]. The results of the present study with respect to healing time demonstrate that the use of GS after the combined procedures does not affect recovery time or the number of dressing changes necessary between any of the groups. The recovery time observed in our study was similar to that reported in other studies with WR alone or combined WR/SP [4,9,25,26,27]. As in previous studies, secondary intention healing using GS has been associated with excellent cosmetic appearance and a high level of patient satisfaction [13]. In any of the experimental groups, complications were not recorded with respect to postoperative infection rate, risk of encapsulation, or tissue reaction.

The available evidence demonstrates that the addition of phenol when performing a partial nail avulsion dramatically reduces symptomatic recurrence, but at the cost of increased postoperative infection [5,28]. As in previous studies, we have observed that the removal of cauterized tissue after SP reduces the risk of infection [7]. In our study, no postoperative infections were observed in either group. In the experimental groups, both types of absorbable gelatin sponges demonstrated a high swelling capacity and a rapid hemostatic capacity to prevent exudate accumulation, which may have reduced the risk of infection. 

In relation to the rest of the secondary outcomes analyzed in the postoperative period, we have not found significant differences between any of the groups. Pain recorded between groups on the first three postoperative days was very similar, close to 5 on the scale on the first day, mild on the second day, and minimal on the third day. Although the pain recorded on the first and second days in experimental Group B was slightly higher than that reported in the control group and in experimental Group C (see Figure 5). Issa et al. found that the intensity and duration of pain was similar, without statistically significant differences between the SP and WR/SP groups, although it was significantly less in the WR/SP group combined treatment than in the WR alone group [3]. In the same way, other studies demonstrate that postoperative pain intensity was similar with SP alone or combined procedures [3,7,9]. 

With the same HPGS used in our experimental group C, some in vitro and vivo studies carried out in the animal model have demonstrated faster re-absorption and a lower inflammatory response than demonstrated by other CGS [18]. However, we have not found significant differences between the experimental groups regarding the average digital circumference before and after the operation, and the same way as in similar studies performed with the same procedure, no significant differences between the groups in postoperative swelling were recorded [25]. Arista et al. found greater inflammation in individuals in the SP group and in those who combined WR/SP with suture application [9]. In our study, wound closure occurred with secondary intention without the use of sutures or approximation strips, and the average inflammation in the control group was very similar to that recorded in the experimental groups.

The recovery time recorded in our study was not conditioned by the use of gelatin sponges. No significant differences were observed with respect to the variable between any of the groups. The number of cures and recovery time was similar (approximately two weeks) to that reported in other studies with WR alone or combined WR/SP [25,26,27,28]. We consider that performing combined SP/WR with gelatin sponges can avoid the delayed healing associated with the cauterized effect on soft tissues and bone of SP alone or combined WR/SP, maintaining the efficacy reported with the combination of both procedures. The gelatin matrix of sponges has the additional advantage of being biocompatible and was demonstrated to be completely resorbed within two weeks. The mean satisfaction reported on a scale of 0–10 by 85% of patients without signs of recurrence was 9.4 ± 0.9 with favorable secondary intention healing, excellent cosmesis, and a high level of patient satisfaction.

Our results indicate a recurrence rate of 7.4% after a mean follow-up of 40.8 months (range, 34–51), of which only 4.4% were symptomatic. These results are similar to those reported by Fulton et al. with combined WR/SP treatment and other studies with SP alone [3,8]; however, they are higher than those recorded in other studies with combined procedures with a recurrence rate of 0.6%, although with a significantly shorter follow-up period than our study [3,4].

In any of the experimental groups, no adverse reactions were reported during the course of the application. Only one patient in experimental group B was excluded because he had a tissue reaction because he did not attend the first treatment appointment at 48 h. Despite its biodegradation, we recommend that when necessary, excess amorphous fibrin and gelatin sponge is partially removed at 48 h, especially when using CGS.

The combined procedure of WR associated with posterior SP using gelatin sponges is a quick and very effective mode of therapy in the surgical treatment of onychocryptosis and is simple, safe, and easy to perform with minimal postoperative morbidity. On the other hand, GS are hemostatic devices that can be of great benefit for patients with acute bleeding, such as that that can occur in nail surgery after the removal of the tourniquet. Gelatin sponges have a neutral pH that makes them suitable for acting as an ideal carrier for drugs and can be very useful combined with antifibronolitic agents in patients undergoing nail surgery with pharmacological drugs that inhibit blood coagulation.

## 5. Conclusions

Absorbable gelatin sponges proved to be effective and safe devices for the control of minor postoperative bleeding associated with combined SP/WR with favorable secondary intention healing, excellent cosmesis, and a high level of patient satisfaction. The local application of gelatin sponges in ingrown nail surgery may result in a slight increase in the acute inflammatory response without significantly affecting healing time, recovery time, or postoperative pain and swelling.

## Figures and Tables

**Figure 1 jcm-11-02420-f001:**
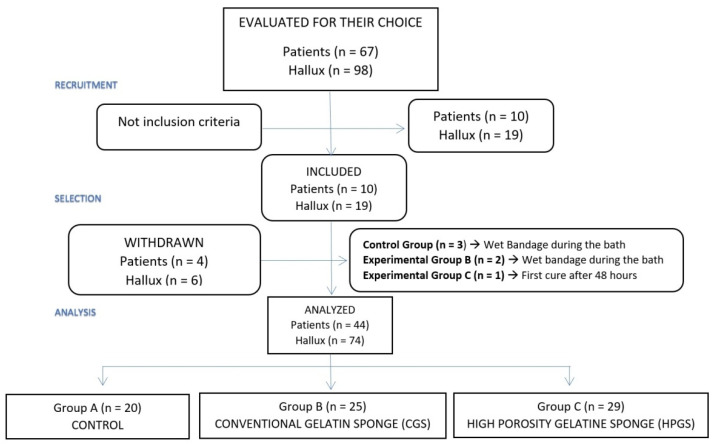
Flow chart of the patients.

**Figure 2 jcm-11-02420-f002:**
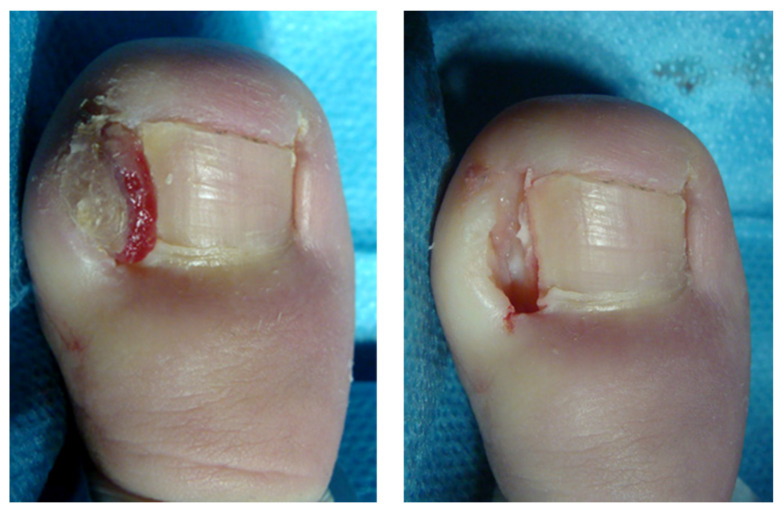
Stage III onychocryptosis in the lateral fold (**left image**). Aspect of the hallux after the Winograd procedure (**right image**).

**Figure 3 jcm-11-02420-f003:**
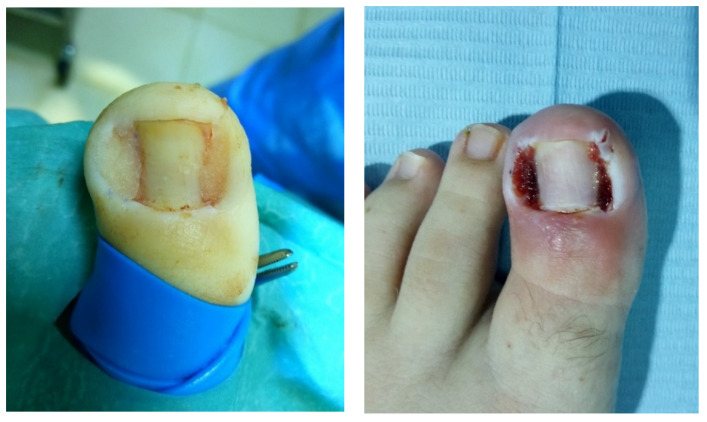
Immediate postoperative aspect (**left**) and 48 h after (**right**) application of gelatin sponges.

**Figure 4 jcm-11-02420-f004:**
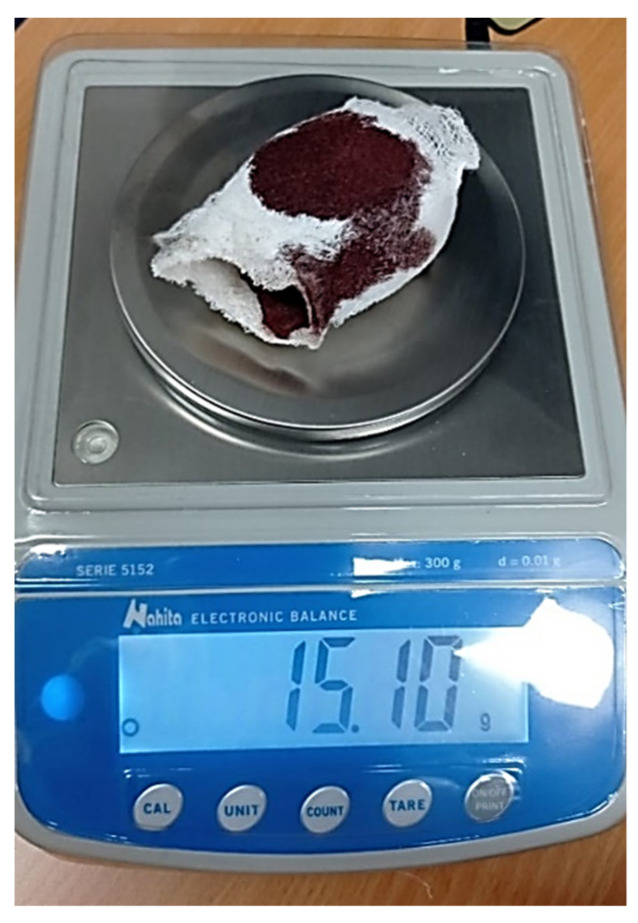
The image shows one of the weighing performed on a bandage of one of the experimental groups.

**Figure 5 jcm-11-02420-f005:**
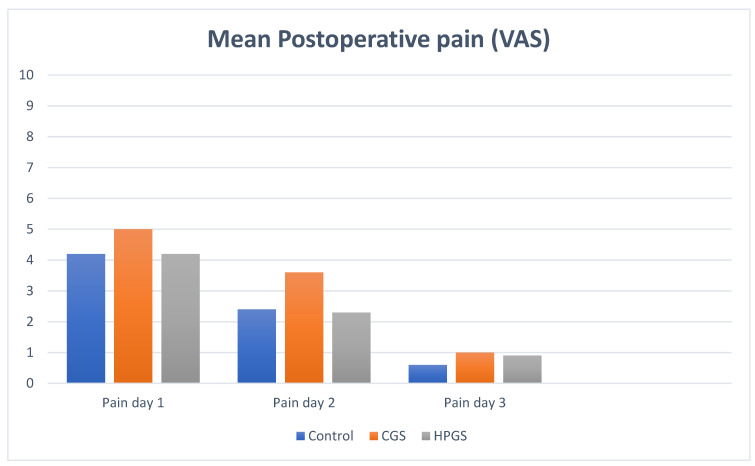
Evolution of the mean pain measured in 3 postoperative days for three groups of treatment. Abbreviations: VAS, visual analog scale; GS, conventional gelatin sponge; HPGS, high porosity gelatin sponge.

**Table 1 jcm-11-02420-t001:** Descriptive characteristics of the study sample.

Characteristics	Group A (Control)	Group B (CGS)	Group C (HPGS)	Total
Average age ± SD (years)	27.4 ± 18.1	30.3 ± 18.3	28.2 ± 18.7	28.7 ± 18.2
Halluces of male	6 (30%)	11 (44%)	10 (34.5%)	27
Halluces of female	14 (70%)	14 (56%)	19 (65.5%)	47
Nail morphology:				
Normal	2 (10%)	4 (16%)	2 (6.9%)	8 (10.8%)
Abnormal	17 (90%)	21 (84%)	27 (93.1%)	66 (89.1%)
Laterality:				
Right	5 (25%)	14 (56%)	20 (69%)	39 (52.7%)
Left	15 (75%)	11 (44%)	9 (31%)	35 (47.2%)
ONC Stage:				
Stage 1	2 (10%)	2 (8%)	2 (6.9%)	6 (8.1%)
Stage 2	1 (5%)	6 (24%)	1 (3.4%)	8 (10.8%)
Stage 3	17 (85%)	17(68%)	26 (89.7%)	60 (81.0%)

Abbreviations: SD, standard deviation; CGS, conventional gelatin sponge; HPGS, high porosity gelatin sponge; ONC, onychocryptosis.

**Table 2 jcm-11-02420-t002:** Outcome measurements in the three treatment groups.

Outcome Measurements	Control (*n* = 20) Mean ± SD (95% CI) Median (IR)	Group CGS (*n* = 25) Mean ± SD (95% CI) Median (IR)	Group HPGS (*n* = 29) Mean ± SD (95% CI) Median (IR)
Post-surgical blood loss (g)	3.5 ± 0.8	1.9 ± 0.8	1.9 ± 0.8
	2.8 (2.3–4.0)	2.1 (1.5–2.6)	1.8 (1.2–2.2)
Digital circumference, pre-operative and at 48 h (cm)	8.3 ± 0.7	8.4 ± 0.7	8.3 ± 0.7
	8.8 (7.9–9.0)	8.1 (8.0–9.1)	8.1 (7.8–8.9)
	8.8 ± 0.6	8.8 ± 0.7	8.8 ± 0.6
	8.9 (8.3–9.2)	8.8 (8.4–9.4)	8.7 (8.3–9.4)
Pain day 1	4.2 ± 1.9	5.0 ± 2.3	4.2 ± 2.5
	4.1 (3.1–5.8)	5.0 (3.0–7.1)	4 (2–6)
	2.4 ± 2.1	3.6 ± 2.7	2.3 ± 2.4
Pain day 2	2.0 (0.3–4.0)	4 (1–6)	0 (0–1.5)
	0.6 ± 1.1	1.0 ± 1.6	0.9 ± 2.0
Pain day 3	0 (0–1.0)	0 (0–1.5)	0 (0–1)
Recovery time (days)	15.1 ± 4.2	15.3 ± 3.6	16.0 ± 4.0
	14 (13–15)	15 (14–20)	14 (14–17.3)
Number of cures	3.0 ± 1.1	3.4 ± 1.0	3.5 ± 1.1
	3 (2–3)	3 (3–4)	3 (3–4)

Abbreviations: SD, standard deviation; CI, confidence interval; IR, interquartile range; CGS, conventional gelatin sponge; HPGS, high porosity gelatin sponge.

**Table 3 jcm-11-02420-t003:** Comparison of the outcome between the control group and the experimental Group B.

Outcome Measurements	Control (*n* = 20) Mean ± SD (95% CI)	Group CGS (*n* = 25) Mean ± SD (95% CI)	*p*-Value
Post-surgical blood loss (g)	3.5 ± 2.3	1.9 ± 0.8	0.005 *
Circumference, pre-operative and at 48 h (cm)	8.3 ± 0.7	8.4 ± 0.7	0.616 **
	8.8 ± 0.6	8.8 ± 0.7	0.979 **
Recovery time (days)	15.1 ± 4.2	16.3 ± 3.6	0.144 *
Number of cures	3.0 ± 1.1	3.4 ± 1.0	0.051 *

Abbreviations: SD, standard deviation; CI, confidence interval; IR, interquartile range; CGS, conventional gelatin sponge. * Kruskal–Wallis test for independent samples. ** One-way ANOVA for independent samples.

**Table 4 jcm-11-02420-t004:** Comparison of the outcome between the control group and the experimental Group C.

Outcome Measurements	Control (*n* = 20) Mean ± SD (95% CI)	Group HPGS (*n* = 29) Mean ± SD (95% CI)	*p*-Value
Post-surgical blood loss (g)	3.5 ± 2.3	1.9 ± 0.8	0.001 *
Circumference, pre-operative and at 48 h (cm)	8.3 ± 0.7	8.3 ± 0.7	0.860 **
	8.8 ± 0.6	8.8 ± 0.6	0.970 **
Recovery time (days)	15.1 ± 4.2	16.0 ± 4.0	0.258 *
Number of cures	3.0 ± 1.1	3.5 ± 1.1	0.094 *

Abbreviations: SD, standard deviation; CI, confidence interval; IR, interquartile range;HPGS, high porosity gelatin sponge. * Kruskal–Wallis test for independent samples. ** One-way ANOVA for independent samples.

**Table 5 jcm-11-02420-t005:** Recurrencein relation to the number of procedures performed.

	Recurrence: No Median (IR)	Recurrence: Yes Median (IR)	*p*-Value
Number of nail folds evaluated (%)	114 (85.07%)	10 (7.4%)	
Follow-up (months)	28.0 ± 20.3	40.8 ± 9.5	0.071 *
	26 (8–50)	39 (34–51)	
Satisfaction scale score (0–10)	9.4 ± 0.9	6.8 ± 3.9	0.004 *
	10 (9–10)	9 (4.5–9.3)	

* Mann–Whitney U-test for independent samples. Abbreviations: SD, standard deviation; CI, confidence interval; IR, interquartile range.

## Data Availability

Please contact acordoba@us.es with any data requests.

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
