# Peer review of "Hemostatic Efficacy of Absorbable Gelatin Sponges for Surgical Nail Matrixectomy after Phenolization—A Blinded Randomized Controlled Trial"

_jcm, 2022, doi:10.3390/jcm11092420_

Round 1
Reviewer 1 Report
The authors described "Hemostatic efficacy of absorbable gelatin sponges for surgical nail matrixectomy after phenolization. A blinded randomized controlled trial". As this study was a prospective randomized trial, this manuscript after precise revision should have a significant impact for potential readers.
I have some recommendations and suggestions to improve this manuscript.
- I believe that Figure 4 and 5 are not necessary because there were not differences significantly.
- If you think they were necessary, when did you measure the circumferences in Figure 5? Please exchange the horizontal content.
- In Table 2, comparisons were confusing. Each p-value is necessary to compare the groups. Please revise the Table 2 to make it easier to understand.
- After all, in the authors experience, both GS and HPGS are effective in reducing bleeding after combining SP/WR. So, the authors should add a Figure regarding bleeding.
- More figure legends in details should be added in Figure 2 (e.g. after surgery,,).
Author Response
Dear reviewer, we appreciate any and all recommendations and suggestions made by you. All the changes included in the manuscript have been highlighted in yellow.
Based on the recommendations, we have the following:
- Replaced figures 4 and 5.
- Tables 3 and 4 have been incorporated to compare the groups more clearly.
- Figure 4 has been incorporated with regard to bleeding.
- We have added more figure legends in detail in Figure 2
Sincerely yours,
Antonio Córdoba-Fernández DPM, PhD
Adrián Lobo-Martín, DPM, MSc

Reviewer 2 Report
The research article entitled “Hemostatic efficacy of absorbable gelatin sponges for surgical nail matrixectomy after phenolization. A blinded randomized controlled trial” revealed an excellent work for the application of gelatin sponges in nail surgery. This paper demonstrated a very good scientific concept for the journal scope. I would like to suggest some points to improve the clarity of your manuscript.
-For the introduction part, the authors should add more information about the application of gelatin sponges in different fields (lines 71-75) to highlight the use of this product for hemostatic efficacy or other properties before you work in nail surgery as follows :
https://pubmed.ncbi.nlm.nih.gov/25496177/
https://pubmed.ncbi.nlm.nih.gov/26112689/
https://doi.org/10.1159/000444320
https://pubmed.ncbi.nlm.nih.gov/24632982/
https://pubmed.ncbi.nlm.nih.gov/23812224/
https://pubmed.ncbi.nlm.nih.gov/22086589/
https://pubmed.ncbi.nlm.nih.gov/25496177/
- For figures 4 and 5 … please add the error bars in the figure. Figure 4 was a better representation than the data in Table 2. Please use the figure 4 with details of the y axis (title, unit, and error bar).
Why figure 5, control and HPGS were combined to compare with GS?
Author Response
Dear reviewer, we appreciate any and all recommendations and suggestions made by you. All the changes included in the manuscript have been highlighted in yellow.
Based on the recommendations, we have the following:
In the Introduction section, we have added 3 more references than those suggested by the reviewer to add more information about the application of gelatin sponges in different fields and highlight the use of these hemostatic devices.
Sincerely yours,
Antonio Córdoba-Fernández DPM, PhD
Adrián Lobo-Martín, DPM, MSc

Round 2
Reviewer 1 Report
The authors revised the article precisely.